# The Potential of Mushroom Extracts to Improve Chemotherapy Efficacy in Cancer Cells: A Systematic Review

**DOI:** 10.3390/cells13060510

**Published:** 2024-03-14

**Authors:** Jéssica Fonseca, Josiana A. Vaz, Sara Ricardo

**Affiliations:** 1UCIBIO—Applied Molecular Biosciences Unit, Toxicologic Pathology Research Laboratory, University Institute of Health Sciences (UCIBIO-IUCS-CESPU), 4585-116 Gandra, Portugal; jessica.fonseca@ipb.pt; 2Associate Laboratory i4HB—Institute for Health and Bioeconomy, University Institute of Health Sciences—CESPU, 4585-116 Gandra, Portugal; 3Centro de Investigação de Montanha (CIMO), Instituto Politécnico de Bragança, Campus de Santa Apolónia, 5300-253 Bragança, Portugal; 4Laboratório Associado para a Sustentabilidade e Tecnologia em Regiões de Montanha (SusTEC), Instituto Politécnico de Bragança, Campus de Santa Apolónia, 5300-253 Bragança, Portugal; 5Differentiation and Cancer Group, Institute for Research and Innovation in Health (i3S), University of Porto, 4099-002 Porto, Portugal

**Keywords:** mushroom extracts, cancer, anticancer therapies, chemoresistance, synergy, drug resistance

## Abstract

Chemoresistance is a challenge in cancer treatment, limiting the effectiveness of chemotherapy. Mushroom extracts have shown potential as treatments for cancer therapies, offering a possible solution to overcome chemoresistance. This systematic review aimed to explore the role of mushroom extracts in enhancing chemotherapy and reversing chemoresistance in cancer cells. We searched the PubMed, Web of Science and Scopus databases, following the PRISMA guidelines, and registered on PROSPERO. The extracts acted by inhibiting the proliferation of cancer cells, as well as enhancing the effect of chemotherapy. The mechanisms by which they acted included regulating anti-apoptotic proteins, inhibiting the JAK2/STAT3 pathway, inhibiting the ERK1/2 pathway, modulating microRNAs and regulating p-glycoprotein. These results highlight the potential of mushroom extracts to modulate multiple mechanisms in order to improve the efficacy of chemotherapy. This work sheds light on the use of mushroom extracts as an aid to chemotherapy to combat chemoresistance. Although studies are limited, the diversity of mushrooms and their bioactive compounds show promising results for innovative strategies to treat cancer more effectively. It is crucial to carry out further studies to better understand the therapeutic potential of mushroom extracts to improve the efficacy of chemotherapy in cancer cells.

## 1. Introduction

Cancer is a health crisis that deeply affects people worldwide. Every year, the number of diagnosed cancer cases continues to rise, surpassing 19 million [1]. This leads to illness and death for individuals. Traditional cancer treatment primarily relies on surgery, radiotherapy and/or chemotherapy [2,3]. Chemotherapy uses drugs that target dividing cells such as cancer cells. Unfortunately, it can result in undesired side effects and a risk of recurrence as well as the development of chemoresistance [4,5,6]. Chemoresistance in cancer is the reduced sensibility or the resistance of cancer cells to the effects of chemotherapy [7]. This can occur due to a decrease in the intracellular concentrations of active agents, alterations in the molecular targets of the drugs, an improvement in the ability to repair the alterations induced by the drugs, an increase in anti-apoptotic mechanisms and a decrease in pro-apoptotic mechanisms [8].

Multidrug resistance (MDR) is the biggest obstacle in chemotherapy and the main cause of treatment failure. MDR encompasses several mechanisms, including genetic and epigenetic alterations that give cancer cells survival advantages in the face of treatment [4,6,9,10,11,12]. MDR can occur due to changes in enzymes that metabolise drugs, proteins being targeted by medication transporters from the ATP binding cassette (ABC) family or modifications in pathways that signal cell death within cancer cells. For example, MDR1, also known as ABCB1 or ATP Binding Cassette Subfamily B Member 1, codes for P glycoprotein (P-gp), an efflux pump responsible for removing various drugs from cells. This protein is linked to resistance to anthracyclines, vinca alkaloids, actinomycin D, Taxol (e.g., paclitaxel) and other substances [3,12,13]. The family of glutathione S transferases also plays a role in MDR by managing the removal of drugs from the cells and inhibiting the mitogen activated protein kinase (MAPK) pathway. This ultimately results in the reduced effectiveness of chemotherapy and an increased ability to resist cell death [6,7]. However, in around 90% of cases, chemotherapy fails due to the tumour cells acquiring resistance [12]. This results in increased invasiveness of the cancer and the formation of metastases, making treatment more complicated. MDR poses a critical challenge in clinical practice, compromising the efficacy of therapies and elevating the risk of cancer recurrence. Hence, there is an imperative need to find strategies capable of reverting MDR-related chemoresistance and, when possible, lowering the chemotherapy basal toxicity.

Natural-based therapeutic strategies are considered to have few side effects. Traditional Chinese medicine has recognised the importance of mushrooms for many years, considering them an alternative for cancer treatment due to their properties and low toxicity [3,14,15,16,17]. Mushroom extracts have already been used to prevent and relieve symptoms in the treatment of both early-stage and advanced cancer treatments [3,16]. Mushroom extracts in combination with chemotherapy drugs have been shown in clinical trials to reduce the side effects of chemotherapy, such as nausea, bone marrow suppression, anaemia and insomnia [6]. Some studies have demonstrated that mushroom extracts can increase the expression of the normal p53 protein, which is involved in regulating and managing tumour development [18,19]. Furthermore, research on cancer cells, like lung carcinoma, breast cancer, colon cancer and adenocarcinoma, has indicated that mushroom extracts can have an impact by slowing down their growth [15]. Combination therapy featuring natural products with chemotherapeutic agents has unveiled a synergistic effect, significantly enhancing tumour cell sensitivity to chemotherapy and surpassing the efficacy of individual drugs [7,9]. Treatments involving breast cancer cells subjected to simultaneous administration of *Suillus collinitus* methanolic extract and etoposide resulted in greater cell growth inhibition than treatment with each component isolated [20]. Moreover, research studies have demonstrated that polysaccharide extracts from *Ganoderma lucidum* inhibit the activation of NF κB by reducing P-gp expression in cancer cells [21]. Combining mushroom extracts with chemotherapeutic agents has been shown to be an effective control approach, preventing the progression of different types of cancer, as well as acting against MDR mechanisms, increasing the effectiveness of chemotherapeutic drugs and playing a sensitising effect on chemoresistant cells [3,6,9,20,21]. This suggests that this combination may not only potentiate the effect of these drugs, but also help to overcome MDR and control tumour growth. Therefore, the main objective of this review is to evaluate and consolidate the existing evidence on how mushroom extracts can reverse chemotherapy resistance when administered in combination. In addition, we explore the molecular mechanisms behind these extracts’ therapeutic effects and their potential synergistic effect with chemotherapy drugs.

## 2. Materials and Methods

### 2.1. Search Strategy

This was a systematic review following the Preferred Reporting Items for Systematic Reviews and Meta-Analyses (PRISMA) guidelines [22]. It is registered in the international PROSPERO database under nº: CRD42023446568.

The reviewers, JF, SR and JV, started searching the Web of Science, Scopus and Pubmed databases on 6 September 2023. The search was adapted to each database, using the following terms: (Mushrooms OR mushroom extracts OR medicinal mushrooms) AND (synergy OR combined effect OR synergistic interaction OR potentiation OR interaction) AND (chemotherapy OR chemotherapeutic agents OR cancer treatment) AND (chemoresistance OR treatment resistance OR drug resistance OR mechanisms of resistance OR treatment). No time limit. We considered articles in English, Spanish, French and Portuguese. The clinical trials study was conducted by JF, SR and JV using the databases of ClinicalTrials.gov and the International Clinical Trials Registry Platform (ICTRP) as of 6 September 2023. No time limit and no filters. In the search on the ClinicalTrials.gov database, it was necessary to adapt the search terms as follows:Condition or disease: (chemotherapy OR chemotherapeutic agents OR cancer treatment).Intervention/Treatment: (mushrooms OR mushroom extracts OR medicinal mushrooms).Other terms: (synergy OR combined effect OR synergistic interaction OR potentiation OR interaction) or (chemoresistance OR treatment resistance OR drug resistance OR mechanisms of resistance OR treatment).

### 2.2. Eligibility Criteria

There were no restrictions on publication date, but for language, only English, Spanish, French and Portuguese were selected, meeting the following criteria. PICOS criteria (participants, intervention, comparison and outcome) were used as described in Table 1.

### 2.3. Inclusion and Exclusion Criteria

As inclusion criteria, we were looking for studies investigating the use of mushroom extracts in combination with anticancer therapies and in vitro or in vivo studies evaluating the effects of mushroom extracts on reducing chemoresistance and providing information on the cellular mechanisms underlying the synergy between mushroom extracts and anticancer therapies.

Regarding clinical trials, we were looking for studies involving human beings. As exclusion criteria, studies that did not investigate the use of mushroom extracts in synergy with anticancer therapies were not included in this review, as well as studies that did not provide relevant information on the reduction of chemoresistance or the cellular mechanisms involved, studies that were not available in full text and studies that were not in one of the previously established languages (English, Spanish, French or Portuguese). Review and meta-analysis studies were also excluded.

### 2.4. Data Extraction and Data Synthesis

The synthesis process was performed in the following steps:Identification: The study selection process began with an examination of the title and abstract information from the Web of Science and PubMed databases, as well as clinical trials from ClinicalTrials.gov and ICTRP. All references were exported to the Rayyan systematic review data management platform (https://www.rayyan.ai) [23]. After, all duplicates were removed. The data were exported to the reference manager.Screening: JF and SR independently analysed the titles and abstracts of the studies, applied the eligibility criteria, and all irrelevant studies were excluded.Eligibility and selection: JF initially chose full-text articles for inclusion, which were then reviewed by SR. If there was any uncertainty about the appropriateness of an article, a third investigator, JV, was brought in to assist.Data extraction: For clinical trials, a table was filled with the following details: registration number, authors’ names, year of registration, location, study setting, study duration, participant characteristics, sample size, study phase, study design, intervention, primary objective, and outcome. There were no specific time limits. We included full-text articles written in English, Spanish, French and Portuguese. In terms of in vivo and in vitro studies, the focus was on selecting those that investigated the use of mushroom extracts combined with anticancer therapies. Data were extracted that provided information on treatment efficacy and the molecular mechanisms underlying the synergy between mushroom extracts and anticancer therapies.

### 2.5. Risk of Bias Assessment

A toxicological data reliability assessment tool (ToxRTool) [24] was used to assess the quality of the in vitro studies included. The in vivo studies were assessed using SYRCLE’s RoB tool for animal intervention studies [25].

Randomised clinical trials were carried out according to the Cochrane risk-of-bias tool for randomised trials (RoB 2) [26] and non-randomised clinical trials were assessed using the ROBINS-I tool (Risk Of Bias In Non-randomized Studies-of Interventions) [27]. According to the scales, the studies were assessed (“low risk”, “high risk” or “unclear risk”).

## 3. Results

### 3.1. Characteristics of the Included Studies

The database search resulted in 1404 results: 1292 in PubMed, 107 in Web of Science and 5 in Scopus. A total of 75 duplicates were excluded, resulting in a final balance of 1329 results. After evaluating the titles and abstracts, 1176 records were excluded: 631 records for being outside the scope of the work; 291 records that, although related to cancer, did not deal with the combination of mushroom extracts with chemotherapy drugs; 229 for being review studies; 19 for being systematic reviews or meta-analyses; 3 for being case reports; 2 for being comments or letters; and 1 for being in a foreign language parallel to those previously established. As a final result, 153 studies proceeded to full-text reading, of which 148 did not meet the inclusion criteria described in Section 2.3, and were excluded, while 5 records met all the criteria and were included in this review (Figure 1).

### 3.2. Assessment of the Risk of Bias

Four of the five studies included in this systematic review carried out in vitro tests, which led us to analyse the quality of these studies using the ToxRTool assessment scale [24] (Figure 2A). Three of the five studies included in this work carried out in vivo studies, which meant that they had to be assessed accordingly, so these two studies were subjected to an assessment of the Risk of Bias using the SYRCLE RoB scale [25] (Figure 2B).

Regarding clinical trials, it was not possible to assess the Risk of Bias in the clinical trials with numbers NCT05763199 and RCT20190822044579N1, as the studies are still ongoing and do not yet have results. The same happened with study number NCT01685489, which was interrupted due to a lack of funding, so it was not possible to assess its results. The Risk of Bias assessment was carried out according to RoB 2 [26] in two clinical trials (NCT00970021 and NCT02486796) (Figure 3A). Clinical trial NCT00779168 was a non-randomised clinical trial and for this reason the quality analysis of this study was judged according to the ROBINS-I tool [27] (Figure 3B).

### 3.3. Study Purpose

In Zhang et al.’s [28] study, the researchers aimed to determine whether the polysaccharide extracted from *Lentinula edodes* (MPSSS) could enhance the effectiveness of docetaxel treatment in prostate cancer by inhibiting chemotherapy resistance induced by cancer-associated fibroblasts (CAFs). CAFs play a crucial role in resistance to prostate cancer treatment, and the researchers aimed to clarify the underlying mechanisms of this resistance. CAFs are described as inherently resistant to docetaxel [28]. Similarly, Gou et al. [29] aimed to investigate whether the use of polysaccharides extracted from the mushroom *Trametes robiniophila* Murr could negatively regulate the expression of microRNAs that target the MDR1 gene and, consequently, regulate the levels of the P-gp protein and thus increase the sensitivity of hepatoma cells to oxaliplatin [29]. Doğan et al. [30] also focused on investigating the action of extracts on P-gp activity using aqueous, methanol, and ethanol extracts from *Fomes fomentarius* and *Tricholoma anatolicum* in breast cancer cells (MCF-7) that were resistant to paclitaxel (PTX) and vincristine due to P-gp overexpression [30]. Xu et al. [31], on the other hand, dedicated their research to investigating the antioxidant effects and potential chemoprophylactic properties of the aqueous extract of *Pleurotus pulmonarius* concerning liver cancer. The study aimed to assess how *Pleurotus pulmonarius* affects the capture of free radicals, including the DPPH radical, superoxide anion radical, hydroxyl radical, and hydrogen peroxide. Additionally, the study examined whether the administration of *Pleurotus pulmonarius* could prevent the development of liver cancer (chemoprophylaxis) and enhance the sensitivity of liver cancer tumour cells to cisplatin treatment. The study also investigated whether *Pleurotus pulmonarius* had potential adverse side effects in mice by assessing critical organs and the animals’ body weight [31]. Cen et al. [32] evaluated the effects of the combination of Sporoderm-Broken Spores of *Ganoderma lucidum* (SBSGL) and Ganoderic Acid D (GAD) with cisplatin in ovarian cancer. The main objective was to determine whether SBSGL and GAD could enhance the effectiveness of cisplatin in ovarian cancer while minimising the harmful side effects associated with cisplatin [32]. All the included studies are summarised in Table 2.

### 3.4. Experimental Methodology and Treatments Used

To achieve their goal, Zhang et al. [28] used two cell lines, CAFs (naturally resistant to docetaxel) and a human prostate cancer cell line, PC-3. Docetaxel was chosen as the chemotherapy. To investigate the effect of MPSSS on prostate CAFs, a conditioned medium derived from CAFs (CAFs-CM) and a conditioned medium of CAFs pre-treated with MPSSS (CAFs-TCM) were first prepared, in which CAFs were treated with MPSSS at concentrations of 0 to 0.25 mg/mL. PC3 cells were then cultured in CAFs-CM and CAFs-TCM for 48 h, followed by treatment with different concentrations of docetaxel for 48 h [28]. The drugs used by Gou et al. [29] were oxaliplatin, a third-generation chemotherapy agent, and *Trametes robiniophila* Murr polysaccharides. They chose to use in vitro and in vivo experimental models to assess the effects of oxaliplatin, *Trametes robiniophila* Murr polysaccharides and the combination of the two. In the in vitro model, they used HepG2 and Bel-7404 human hepatocellular carcinoma cells and treated them with the drugs and/or *Trametes robiniophila* Murr polysaccharides for 24, 48 and 72 h. Bel-7404 cells were treated with doses of 0.27 mM *Trametes robiniophila* Murr polysaccharides, 10.08 μM oxaliplatin or 0.27 mM *Trametes robiniophila* Murr polysaccharides and 10.08 μM oxaliplatin. HepG2 cells were treated with doses of 0.20 mM *Trametes robiniophila* Murr polysaccharides, 5.04 μM oxaliplatin, or 2.0 mM *Trametes robiniophila* Murr polysaccharides and 5.04 μM oxaliplatin. They also created a nude mouse xenograft model (BALB/c) inoculating HepG2 and Bel-7404 cells to evaluate the effects in vivo. The mice were injected subcutaneously with 0.1 mL of HepG2 and Bel-7404 cells at a concentration of 1.0 × 107/mL. When the tumour volume reached 90–100 mm^3^, the mice were randomly divided into groups (*n* = 6/group) and treated for 21 days. The *Trametes robiniophila* Murr group received 2.6 g/kg body weight of *Trametes robiniophila* Murr polysaccharides by gavage once a day, the oxaliplatin group was injected intraperitoneally with 10 mg/kg body weight once a week, and those in the *Trametes robiniophila* Murr and oxaliplatin group received 2.6 g/kg body weight of *Trametes robiniophila* Murr by gavage once a day and were injected intraperitoneally with 10 mg/kg body weight of oxaliplatin once a week. The control group was treated with an equivalent volume of normal saline solution once a day [29].

Doğan et al. [30] used the chemotherapy drugs PTX and vincristine in the MCF 7 breast cancer cell line. Sublines resistant to PTX (MCF-7/Pac) and vincristine (MCF-7/Vinc) were developed from drug-sensitive MCF-7/S cells to reproduce chemoresistance in this type of cancer and thus analyse whether the extracts of *Fomes fomentarius* and *Tricholoma anatolicum* combined with chemotherapeutics were able to regulate P-gp activity and increase the cells’ sensitivity to the drugs. The MCF-7/Pac and MCF-7/Vinc cell lines have been described as having an overexpression of the MDR1 gene [30]. Although Xu et al. [31] did not focus entirely on evaluating chemoresistance, an in vivo model was used for this purpose. The main focus of this work was to evaluate the potential of *Pleurotus pulmonarius* extracts in chemoprophylaxis, trying to understand whether prior oral administration of *Pleurotus pulmonarius* could prevent cancer by inhibiting the proliferation of cancer cells (Huh7 liver cancer cells). With regard to assessing chemoresistance, they sought answers on the chemosensitising effect of *Pleurotus pulmonarius* extracts and, to this end, used an in vivo model (nude mice). Liver cancer xenotransplants were established by injecting 5 million Huh7 cells into the flanks of mice and after three days the mice were separated into four groups. Group 1 was given a dosage of 2 mg/kg of cisplatin every three days through an injection in their abdomen. Group 2 received a dose of *Pleurotus pulmonarius* orally. It was not strong enough to inhibit any effects. Group 3 received a combination of cisplatin (injected every three days) together with daily oral *Pleurotus pulmonarius*. Lastly, Group 4 served as the control group, receiving only the vehicle. The experiment lasted 28 days, with continuous monitoring of the animals’ body weight [31].

Similar to Xu et al. [31], Cen et al. [32] used cisplatin as a chemotherapy agent. The ovarian cancer SKOV3 and SKOV3/DDP (cisplatin-resistant) cell lines were used in in vitro and in vivo models. Tumours were established by injecting 4 × 10^6^ SKOV3 cells into each mouse, and when the tumours reached approximately 100 mm^3^, the animals were randomly divided into four groups (*n* = 5 in each group) and treated with different combinations of cisplatin and SBSGL: (I) in the SBSGL group, mice underwent daily oral administration of SBSGL at a dose of 2 g/kg; (II) in the cisplatin group, mice were given cisplatin injections once a week at a dosage of 3 mg/kg; (III) in the combination group, mice were first treated with SBSGL for three days and then received SBSGL and cisplatin following the previously mentioned schedules. In the control group, mice received daily oral administration and weekly intraperitoneal injections of an equivalent volume of saline solution, as per the other groups. Parameters such as animal weight, tumour volume and weight and histological analyses were carried out to evaluate proliferation and apoptosis. Other organs, such as the kidneys, intestines, bone marrow and peripheral blood, were histologically evaluated to measure the impact of the treatment on neighbouring tissues [32].

### 3.5. Mushroom Extract Production

In their study, Zhang et al. [28] successfully extracted a polysaccharide called MPSSS (577.2 KD) from *Lentinula edodes* mushroom. The extraction process involved dissolving the mushroom extract in deionised water and then combining it with ethanol in a 1:1 ratio. After centrifugation, the resulting precipitate was dissolved in a 30% ethanol solution and stored at 4 °C. Further centrifugation allowed for the collection of the precipitate, which was then dissolved in ethanol to create 40%, 50% and 80% ethanol solutions using the method. This process resulted in a powder known as MPSSS, which consists of glucose, galactose and mannose [28]. The team of Gou et al. [29] used polysaccharides of *Trametes robiniophila* Murr obtained commercially [29].

Doğan et al. [30] were the only researchers to describe collection. Harvesting took place in the Konya–Seydişehir and Karaman–Ermenek–Başyayla regions in Tukey between October and November. The samples were dried at 37 °C for around 3 to 5 days and pulverised by milling. The extraction process was carried out using an ultrasonic homogeniser in solvents (ultrapure water, methanol and ethanol) after pulverisation. The ultrasound temperature was kept between 25 °C and 37 °C, and the frequency was 20 kHz. Each 15 g mushroom powder sample was extracted in 200 mL of solvent using ultrasound for 60 min. The extract suspension of the samples was centrifuged (at 8500 rpm, 15 min, 4 °C), and the supernatant of the samples was collected; this method was repeated twice until the extracts were completely dissolved. The extracts were then filtered to remove particles and debris. The solvent from the filtered extract solution was evaporated using a rotary evaporator at 35–40 °C and 200–250 rpm to remove the methanol and ethanol. The extracts were stored at −86 °C and lyophilised at −110 °C to obtain dry samples [30].

Xu et al. [31] purchased samples of *Pleurotus pulmonarius* mushrooms from the local market. The freeze-dried mushrooms were ground into powder and extracted with hot water [sample–solvent ratio 1:25 (*w/v*)] at room temperature (25 °C) for 3 h. After extraction, the insoluble residue was removed by centrifugation at 4000× *g* for 5 min. The supernatants were frozen to obtain the hot water-soluble mushroom extract. The characterisation of *Pleurotus pulmonarius* metabolites was described in previous work by the group and it was found that the main bioactive compound was a polysaccharide–protein complex [31]. Cen et al. [32] used a sample of *Ganoderma lucidum* and, to prepare the extract, the intact *Ganoderma lucidum* spores were ground in a supersonic air blast to break down the Sporoderm and then extracted by water twice in volumes of ten and eight times. The combined solution was filtered, concentrated and dried to obtain SBSGL, whose essential composition is the triterpene GAD. The yield of SBSGL was approximately 10 percent by mass. As for the preparation of the test solution used in the animal experiment, the SBSGL was reconstituted in ddH2O at 0.2 g/mL and the mixture was heated in boiling water for 20 min. As it was a suspension, the SBSGL was immediately mixed upside down and aliquoted. The stock was stored at −20 °C and used within a week [32].

### 3.6. Anti-Proliferative and Cytotoxic Activity

In the study by Zhang et al. [28], the results initially showed that the percentage of dead cells was lower when PC3 cells were cultured with conditioned medium derived from CAFs, suggesting that some components of this medium protected the cells from docetaxel-induced death. However, PC3 cells cultured in CAF-conditioned medium treated with MPSSS showed a more marked decrease in viability compared to cells cultured in CAF-conditioned medium not treated with MPSSS. This suggests that CAFs play an important role in promoting cell proliferation or inhibiting tumour cell death but that the protective role of CAFs was weakened by pre-treatment with 0.5 mg/mL MPSSS, increasing the sensitivity of CAFs to docetaxel. In additional experiments, it was found that CAFs are naturally resistant to docetaxel up to a concentration of 240 nM, but when treated with 0.5 mg/mL MPSSS, the sensitivity of CAFs to docetaxel increased [28].

In the study carried out by Gou et al. [29], in vitro, *Trametes robiniophila* Murr polysaccharides were able to significantly inhibit the viability of human HepG2 and Bel-7404 cells in a dose-dependent manner at 48 and 72 h. They also showed that when *Trametes robiniophila* Murr polysaccharides were combined with oxaliplatin they had a greater inhibitory effect than the drug alone. The clone formation assay showed that the number of clones in the combined treatment group was significantly lower than that in the group treated with oxaliplatin alone, indicating greater inhibition of cell proliferation in the group receiving the combined treatment compared to the oxaliplatin-alone group. In vivo, the results indicated that the xenograft volume of the inoculated nude mice (Bel-7404 cells and HepG2 cells) was significantly lower in the group receiving the combined treatment than in the oxaliplatin group after 10 or 13 days of drug administration. This confirms the anti-proliferative properties of the combination of *Trametes robiniophila* Murr polysaccharides and oxaliplatin [29].

Doğan et al. [30] observed varying levels of efficacy of the extracts in inhibiting cell growth. In the *F. fomentarius* extracts, the IC_50_ value for the aqueous extract was 0.76 mg/mL, for the methanolic extract was 0.35 mg/mL and for the ethanolic extract was 1.08 mg/mL. In the *T. anatolicum* extracts, the IC_50_ values were 4.28 mg/mL for the aqueous extract, 4.98 mg/mL for the methanolic extract and 2.22 mg/mL for the ethanolic extract. Based on these results, it was found that the aqueous extracts of *F. fomentarius* were more effective on the cells than the methanol and ethanol extracts, which showed no significant anti-proliferative effects on the drug-resistant cell lines [30]. Xu et al. [31] did not specify the activity of the extracts they studied. However, it was observed that *Pleurotus pulmonarius* alone inhibited tumour growth by 22.6%, while cisplatin alone inhibited it by 35.5%. When administered together, *Pleurotus pulmonarius* and cisplatin had a synergistic effect resulting in a remarkable 88.5% inhibition rate. These findings suggest that the simultaneous use of *Pleurotus pulmonarius* and cisplatin significantly enhances the anti-proliferative effects of chemotherapy. Therefore, the combination treatment is highly effective in reducing cancer cell proliferation [31].

Cen et al. [32] showed that both cisplatin and SBSGL exhibited significant cytotoxic and anti-proliferative activities in SKOV3 and SKOV3/DDP ovarian cancer cell lines. The administration of cisplatin was conducted at concentrations of 40 μM for SKOV3 and 200 μM for SKOV3/DDP, while SBSGL was given at a concentration of 200 μM. Regarding tumour volume and weight, the group that received the combination of cisplatin and SBSGL showed more pronounced inhibitory effects on tumour growth compared to the group that received treatment with cisplatin alone. In addition, the results showed that the tumours in the combined treatment group had a significantly reduced Ki-67 level (a marker of the cell proliferation rate), as well as a higher TUNEL (indicative of apoptosis) fluorescence intensity, suggesting that the combination of SBSGL with cisplatin further suppressed proliferation and promoted apoptosis of ovarian tumours in vivo [32].

To clarify how SBSGL could sensitise ovarian tumours to cisplatin, they used GAD, which reduced the viability of SKOV3 and SKOV3/DDP cells as a function of time and concentration. After 24 h of treatment, the IC_50_ values were 39.917 μM in SKOV3 cells and 207.191 μM in SKOV3/DDP cells. To explore whether GAD could increase the cytotoxicity of cisplatin, the non-toxic concentration of GAD (200 μM for both cell lines) and cisplatin at IC_50_ values of 40 μM for SKOV3 and 200 μM for SKOV3/DDP were used for 24 h. According to the results, this combination further reduced cell viability in both cell lines, and this inhibitory effect was positively related to the concentration of GAD, demonstrating that GAD potentiated the inhibition of proliferation induced by cisplatin in both cell lines. In addition, the colony formation assay was carried out and the results showed that, compared to treatment with cisplatin alone, the combination of GAD with cisplatin was shown to further inhibit the rate of colony formation in both cell lines. When investigating cell death, the rate of apoptosis in SKOV3 cells in the treatment with GAD and cisplatin (12.82%) was four times lower in the treatment with cisplatin alone (3.19%), and these values followed the same trend in SKOV3/DDP cells. These findings suggest that these substances effectively suppressed the growth of cancer cells and triggered cell death. Moreover, SBSGL seemed to amplify the effects of cisplatin, leading to a combination treatment compared to using cisplatin alone [32].

### 3.7. Reversal of Chemoresistance and Increased Sensitivity to Chemotherapy

In Zhang et al. [28], MPSSS was able to reverse the resistance of prostate cancer cells (PC3) to docetaxel treatment induced by CAFs. Initially, CAFs suppressed apoptosis in PC3 cells in response to docetaxel treatment, making the PC3 cells resistant to the drug. However, when the CAFs were exposed to MPSSS, this substance effectively counteracted the inhibition of programmed cell death. As a result, PC3 cells treated with MPSSS reacquire sensitivity to docetaxel. The restoration of sensitivity in PC3 cells was induced by CAFs treated with MPSSS. Additionally, the study found that CAFs trigger a process called epithelial–mesenchymal transition (EMT) in PC3 cells, which is associated with drug resistance. However, MPSSS was found to prevent this transition, indicating that it may also inhibit the ability of CAFs to promote docetaxel resistance by inducing EMT in tumour cells [28].

Gou et al. [29], when analysing the effects of the combination of polysaccharides from *Trametes robiniophila* Murr with the drug oxaliplatin in HepG2 and Bel-7404 cells, were able to demonstrate that polysaccharides from *Trametes robiniophila* Murr had a strong ability to reduce the viability and growth rate of these cancer cells, and the combination of polysaccharides from *Trametes robiniophila* Murr with oxaliplatin had a more pronounced inhibitory effect than using oxaliplatin alone. Additionally, the study revealed that the expression of the P-gp protein associated with drug resistance decreased in cells treated with *Trametes robiniophila* Murr polysaccharides, both in vivo and in vitro [29]. The results of the study by Doğan et al. [30] confirmed the efficacy of the extracts of *F. fomentarius* and *T. anatolicum*. Assays using Rhodamine 123, a substrate of the P-gp protein, were conducted. In resistant MCF-7/Pac and MCF-7/Vinc cells, Rhodamine 123 was typically pumped out of the cells by P-gp, resulting in low fluorescence. However, when methanol extracts from *F. fomentarius* and ethanol extracts from *T. anatolicum* were added to the resistant cells, there was a significant inhibition of P-gp activity, and Rhodamine 123 was not pumped out of the cells. This resulted in high fluorescence, indicating the reversal of resistance to chemotherapy drugs [30]. In Xu et al. [31], *Pleurotus pulmonarius* extract was shown to sensitise tumour xenotransplants to conventional chemotherapy drugs in vivo. In these mice models, the combination of *Pleurotus pulmonarius* and cisplatin exhibited a synergistic effect. While treatment with cisplatin or *Pleurotus pulmonarius* alone resulted in limited inhibition of tumour growth (35.5% and 22.6%, respectively), the combined therapy achieved a remarkable inhibition rate of 88.5%. Furthermore, the cellular consumption of *Pleurotus pulmonarius* did not result in significant side effects, as evidenced by the histological analysis of critical organs and the monitoring of the mice’s body weight [31]. Cen et al. [32] demonstrated that SBSGL was able to sensitise ovarian cancer to cisplatin, both in in vivo and in vitro experiments. In mouse ovarian tumour models, the combination of cisplatin and SBSGL more effectively inhibited tumour growth than cisplatin alone, without affecting mouse weight loss. In the in vitro tests, a substance called Ganoderic Acid D (GAD), which is the main component of SBSGL, was evaluated alone and was shown to decrease the survival of ovarian tumour cells. When GAD was combined with cisplatin, the inhibitory effect was even greater, and it was observed that its effectiveness was dose-dependent. The decrease in viability was mediated by apoptosis and necrosis. These results highlight the potential of SBSGL to improve the effectiveness of cisplatin treatment in ovarian cancer [32].

### 3.8. Cellular Mechanism Underlying Mushroom Extracts’ Properties

Zhang et al. [28] demonstrated that CAFs increase the expression of anti-apoptotic proteins such as Bcl-2 and reduce the expression of caspase-3, a key protein in apoptosis. TGF-β1, secreted by CAFs, is critical in docetaxel resistance. MPSSS reduces the concentration of TGF-β1, preventing docetaxel resistance. MPSSS acts through the TLR4 receptor on CAFs, inhibiting the JAK2/STAT3 pathway, which is linked to CAF activity and TGF-β1 secretion. This deactivates CAFs and reverses docetaxel resistance in prostate cancer cells [28].

In their research, Gou et al. [29] investigated the mechanisms by which the *Trametes robiniophila* Murr polysaccharide exerted its effects. A crucial part of this mechanism involves the regulation of microRNAs, namely miR-224-5p, which regulates the expression of the ABCB1 gene. *Trametes robiniophila* Murr polysaccharides positively regulated miR-224-5p, which inhibits the expression of the ABCB1 gene and, consequently, P-gp. This increases the sensitivity of hepatoma cells to treatment and thus potentiates the inhibitory effect of oxaliplatin [29]. Similar to Gou et al. [29], in Doğan et al. [30] the mechanism of action by which *F. fomentarius* and *T. anatolicum* extracts were shown to reverse MDR and increase chemotherapy sensitivity involved the inhibition of P-gp protein activity in drug-resistant breast cancer cells [30]. However, in this article the authors do not describe the cellular mechanisms that led to the overexpression of P-gp.

Cen et al. [32] found that GAD affects sensitivity to cisplatin by inhibiting extracellular signal-regulated kinase (ERK1/2) signalling in the mitogen-activated protein kinase (MAPK) pathway. Firstly, GAD promotes the production of reactive oxygen species (ROS), which lead to oxidative stress in tumour cells. The increase in ROS levels is closely linked to sensitivity to cisplatin, since this chemotherapeutic exerts part of its cytotoxicity through the accumulation of ROS. GAD therefore potentiates this mechanism by increasing intracellular ROS levels in tumour cells. This oxidative stress is a factor that negatively affects the activation of ERK1/2 in the MAPK pathway. As ERK1/2 activation is known to play an important role in cisplatin resistance, its activation is necessary for tumour cell survival. However, by inhibiting ERK1/2 activation, GAD weakens the ability of tumour cells to survive and resist treatment with cisplatin. Thus, the combination of GAD with cisplatin triggers a series of events, including an increase in intracellular ROS and the inhibition of ERK1/2, which results in greater sensitivity of tumour cells to cisplatin. The team confirmed this mechanism by reducing the level of the p-ERK antibody in the combined treatment group, and this could be restored when treatment with N-acetylcysteine (an ROS scavenger) was carried out. Finally, GAD inhibits the MAPK pathway by inhibiting ERK1/2, which in turn regulates cellular processes such as cell growth, differentiation and programmed cell death [32]. The mechanism by which *Pleurotus pulmonarius* extract sensitised tumour xenotransplants to conventional chemotherapy drug treatment was not explored in Xu et al.’s [31] study. The cellular mechanisms by which mushroom extracts improved the efficacy of chemotherapy in the studies included in this article are summarised graphically Figure 4.

### 3.9. Clinical Trials

As well as deciphering the cellular mechanisms underlying the properties of mushroom extracts, we also sought to select clinical studies that used mushroom extracts as a complement to cancer treatment, as a strategy to overcome chemoresistance. In the context of cancer, we searched only for the term “cancer” as a condition, and the result showed us that there are currently more than 23,700 studies in the Clinicaltrial.com database and more than 26,500 in ICTRP just in the recruitment phase. After searching the same databases with additional terms as previously described in Section 2.1 of this paper, we obtained a result of 8 clinical trials in the Clinicaltrial.com database and 38 in ICTRP. Although none of the studies found in our research directly evaluated the effects of mushroom extracts as a strategy for increasing the effect of chemotherapy and overcoming chemoresistance, seven clinical trials were identified that used mushroom-derived compounds in their research on cancer patients undergoing chemotherapeutic treatment. In addition to deciphering the cellular mechanism behind mushroom extracts’ properties, we also aimed to select clinical studies using mushroom extracts as a complement to cancer treatment, as a strategy to increase the effect of chemotherapy. Although none of the studies found in our research directly assessed the effects of mushroom extracts in this context, clinical trials using mushroom-derived compounds in their research were identified (Table 3).

The clinical study with reference number NCT00970021, which began in 2009, was carried out to evaluate the effects of an 82% *Agaricus blazei* Murril mushroom extract (Andosan™) as a complementary treatment in multiple myeloma patients undergoing high-dose chemotherapy. The study involved 40 patients divided into two groups, with one group (*n* = 19) receiving the Andosan™ extract and the other group (*n* = 21) receiving a placebo, both in conjunction with chemotherapy. Treatment began with stem cell mobilising chemotherapy and continued until the end of aplasia after high-dose chemotherapy, totalling around seven weeks. The results showed that *Agaricus* extract successfully inhibited the development of myeloma cells in a controlled laboratory environment and that this impact was influenced by the amount administered. The researchers reported improvements in the immune system in patients who received Andosan™, with higher percentages of plasmacytoid dendritic cells (CD303+) in the leukapheresis product collected after stem cell mobilisation, indicating the positive stimulatory effect of the treatment. Additionally, they found higher serum levels of cytokines such as interleukin (IL) 1ra (associated with a better prognosis), IL 5 and IL 7 (can be interpreted as a positive effect of the treatment) compared to those who received a placebo. But, they also noted high serum levels of regulatory T cells (CD4+, CD127d+ and CD25+), which could be interpreted as an immunosuppressive factor with a negative impact.

They also analysed gene expression in eight patients in the *Agaricus* group and six patients in the placebo group. The results showed that in the *Agaricus* group there was an increase in the expression of immunoglobulin genes (IGKC, IgHV4-31 and IGKC) and immunoglobulin killer receptor genes (KIR2DL3 and KIR2DL4), suggesting an immunomodulatory effect of AndoSan™. The increased expression of genes from the human leukocyte antigen (HLA) system indicates that AndoSan™ may also have a direct anti-proliferative effect on myeloma cells, since the control group had a low level of expression of these genes [33].

The clinical trial NCT00779168, started in 2009, evaluated the use of *Agaricus bisporus* mushroom extract in patients with recurrent prostate cancer after local treatment. Patients were given mushroom tablets that were taken twice a day until prostate-specific antigen (PSA) progression, clinical progression or toxicity. The treatment lasted 28 days and the groups were divided according to the dose escalation, which was carried out in cohorts of six patients, each with six dosages: 4, 6, 8, 10, 12 and 14 g/day. If no dose-limiting toxicities were detected during the first 28 days of treatment, the next higher dose (up to 14 g/day) was tested.

A complete response (defined as a decrease in PSA to ≤0.04 ng/mL) of PSA was confirmed after at least 4 weeks of treatment and 36% of patients showed a decrease in PSA levels during the study. The results showed that 11% of participants achieved a complete response; two patients showed a lasting complete response, which was maintained until the date of publication of the results; two patients showed a partial PSA response (reduction of 50% of baseline PSA), one of them with a lasting response; and five patients maintained stable PSA levels. On average, treatment lasted 10 months with minimal side effects, except for one patient who interrupted treatment due to side effects related to severe hyponatraemia, but who was not considered to have dose-limiting toxicities. Cytokine analysis indicated that the IL 15 levels of patients who responded completely to treatment were significantly higher than those of non-responders, suggesting that IL 15 may promote tumour suppression in prostate cancer. In addition, the analysis of peripheral blood revealed a notable decrease in myeloid-derived suppressor cells (CD33+HLA-DR-) in patients who had complete or partial responses after 13 weeks of treatment with mushroom extract. These cells secrete angiogenic growth factors and support tumour metastasis, playing an important role in angiogenesis. The decrease in the levels of these cells supports the potential anti-tumoural role of the extracts [34].

The clinical trial NCT04519879, started in 2021, is a phase 2 study that aims to investigate the effectiveness of white mushroom (*Agaricus bisporus*) supplements in reducing PSA levels in patients with recurrent or favourable-risk prostate cancer without previous treatment. The study involves two groups, one receiving the supplement and one not. Patients are monitored for 48 weeks to assess PSA levels, adverse events and parameters such as peripheral blood mononuclear cells, cytokines, growth factors and chemokines, including IL 15 in plasma. In addition, the aim is to explore immunomodulatory effects, the influence on sexual function and changes in cancer signalling pathways. The study is ongoing and still in the recruitment phase; no results are available.

The NCT02486796 trial studied the impact of treatment with a combination of *Reishi* mushroom extract, coenzyme Q10 and melatonin alongside chemotherapy for breast cancer patients. The study spanned from February 2016 to March 2017 and a total of five participants were included. Its main objective was to assess therapeutic response and quality of life parameters. The results showed no mortality or serious adverse events related to natural-based treatment. Since the number of participants was small, extensive studies are required to draw conclusions.

The ongoing phase 2 pilot study under reference number NCT05763199, started in 2023, aims to evaluate the use of *Lentinula Edodes* Mycelia (AHCC^®^) extract in women with ovarian cancer during adjuvant chemotherapy. It involves 20 participants divided into two groups, one receiving AHCC^®^ in capsules and the other receiving a placebo. The primary objective of the study is to explore how AHCC^®^ impacts the quality of life of ovarian cancer patients. Currently, no participants are being recruited for this study; therefore, no results are available.

Clinical trial NCT01685489, started in 2013, assessed the safety and effectiveness of a treatment for castration prostate cancer. The trial intended to investigate the combination of PSK^®^ (*Krestin* polysaccharide) treatment with docetaxel, a chemotherapy drug. The main objectives were to determine the dosage of PSK^®^ in conjunction with docetaxel as well as to conduct pharmacokinetic analyses. Unfortunately, due to funding this study never took place.

Another ongoing clinical trial, RCT20190822044579N1, started in 2021, aims to examine how consuming probiotic ice cream containing β (*Agaricus bisporus*) may impact breast cancer patients who underwent chemotherapy and radiotherapy treatments. This study involves four groups, three receiving formulations of ice cream and one receiving a placebo ice cream. Although participant recruitment is closed, no results are yet available.

## 4. Discussion

Cancer cells develop characteristics known as “hallmarks of cancer” which contribute to the growth and progression of tumours. These hallmarks include avoiding growth suppression, resisting cell death, promoting excessive cell signalling, stimulating blood vessel formation, enabling replication, invading surrounding tissues, disrupting metabolic processes, evading the system responses triggering pro-tumour inflammation, and causing genomic instability [35]. Recently, four new characteristics have been proposed as significant additions to our understanding of cancer and its microenvironment. These features include epigenetic alterations, cellular regression, the role of microorganisms, and signalling [36]. One of the major contributors to therapy failure is the development of multidrug resistance (MDR). MDR incorporates a variety of mechanisms that give cancer cells a survival advantage against cancer therapies [4,6,9,10,11,12].

Currently, medical priorities are centred on finding new therapeutic strategies with fewer side effects. Mushrooms have gained attention for their health benefits, attributed to their antibacterial, anti-inflammatory, immune-boosting, cancer-fighting, liver-protecting and diabetes-regulating properties [37]. Over the years, several species of mushrooms have already been thoroughly characterised, resulting in the successful identification and isolation of their bioactive compounds [38]. Thanks to decades of research dedicated to understanding the health benefits of mushrooms, we now recognise that their positive effects on health are attributed to the various bioactive compounds present in this type of fungus, such as phytosterol (ergosterol and fungisterol), polysaccharides (e.g., β-D-glucan), statins (competitively blocks 3-hydroxy-3-methyl-glutaryl CoA reductase), glycoproteins, triterpenoids, and flavonoids [38,39,40]. Mushrooms are known for their ability to fight cancer by affecting the cells of the system. These include stem cells, lymphocytes, macrophages, T cells, dendritic cells and natural killer cells. The immunomodulatory properties of mushroom polysaccharides are primarily responsible for these effects. They influence cytokine production, stimulate cells, enhance antibody responses and regulate activity. Ultimately, these polysaccharides contribute to strengthening the tumour immune response [2,41].

With regard to the health benefits of mushrooms, in one study, an ethanolic extract of *Pleurotus pulmonarius* proved effective in reducing the negative effects of a high-fat diet in hyperlipidaemic rats, showing hypolipidaemic, antioxidant and anti-inflammatory properties, inhibiting the levels of NF-κB2, STAT3 and CREB1 [42]. In another study, Gil-Ramírez et al. [43] analysed various parts of *Agaricus bisporus* and parts generally discarded by the mushroom marketing industry, which showed significantly higher inhibitory properties compared to the commercialised parts. The aqueous methanolic extracts of this mushroom were able to inhibit up to 60% of the activity of 3-hydroxy-3-methyl-glutaryl CoA reductase (HMG-CoA reductase) in vitro, which is a key enzyme in the cholesterol biosynthetic pathway [43]. These hypolipidaemic properties of mushroom extracts can be very important in the context of cancer, since cholesterol synthesis is an important signalling pathway for the survival of cancer cells. HMG-CoA reductase is considered a metabolic oncogene that can promote tumour growth and cooperates with Ras for colony formation [44]. The inhibition of HMG-CoA reductase affects the production of isoprenoids, which are essential for the functioning of tumour cells and contribute to resistance and metastasis [44,45,46,47].

The main objective of this study was to review the potential of combining mushroom extracts with anticancer treatments as a strategy for improving therapy results. In this systematic review, we selected a series of articles, partly due to the scarcity of research on this interaction between mushroom extracts and chemotherapeutic drugs. Most studies in this field tend to focus on evaluating isolated compounds extracted from mushrooms, rather than examining the overall extract itself. For example, a study using grifolin (a secondary metabolic product isolated from the *Albatrellus confluence* mushroom) on human ovarian cancer cell lines (A2780) showed that this compound was capable of suppressing cell viability, inducing apoptosis and cell cycle arrest. It deregulated cell cycle proteins and increased the Bax/Bcl-2 ratio (apoptosis regulator), the expression of cleaved caspase-3 and cleaved (ADP-ribose). The expression levels of protein kinase B (Akt) and p-ERK1/2 were significantly reduced in cells treated with grifolin, suggesting that the main mechanism underlying the action of this compound on A2780 cells occurred via the Akt and ERK1/2 signalling pathways [48]. In another study, the authors used 4-acetylanthroquinonol B (a bioactive isolate from the mycelia of *Antrodia camphorata*) on human colon carcinoma cell lines derived from different tumours (DLD-1, HCT-116, SW-480, HT-29 and RKO cells). They compared its anticancer therapeutic potential with that of anthroquinonol and with the combined therapeutic strategy generally used in colon carcinoma (i.e., folinic acid, fluorouracil and oxaliplatin). The isolated compound inhibited tumour proliferation, suppressed tumour growth and reduced chemoresistance, negatively regulating oncogenic signalling pathways important for stem cell maintenance, including Lgr5/Wnt/β-catenin, JAK-STAT and receptor tyrosine kinase, as well as inducing the downregulation of aldehyde dehydrogenase and other stem cell-related factors. The compound’s inhibitory effect on tumour proliferation, its suppression of tumour growth and its attenuation of stem cell-related chemoresistance was more effective when compared to other chemotherapeutic strategies [49]. These results, despite having been achieved using an isolated compound, are in line with the results of the study included in this work and presented by Zhang et al. [28], where a polysaccharide extracted from *Lentinula edodes* was able to negatively regulate the JAK/STAT oncogenic signalling pathway [28].

Cordycepin (isolated from the *Cordyceps militaris* mushroom) is a component found in dietary mushrooms and was used on cancer cell lines (KB-vin, HepG2-vin and NCI-H460/MX20) that showed a multidrug resistance profile. Treatment of the cells with cordycepin was able to modulate P-gp expression and inhibit its function by stimulating P-gp ATPase activity in cells stably expressing P-gp (ABCB1/Flp-InTM-293) and in cancer cells with a multidrug resistance profile (KB-vin). It was also able to significantly re-sensitise MDR cancer cells (KB-vin, HepG2-vin and NCI-H460/MX20) to conventional chemotherapeutic agents (i.e., paclitaxel and mitoxantrone) [50]. This result is in synergy with the results presented by Doğan et al. [30], who described that the extracts of *F. fomentarius* and *T. anatolicum* were able to reverse MDR, increasing sensitivity to chemotherapy by inhibiting the activity of the P-gp protein in drug-resistant breast cancer cells [30]. Also similar are the results of Gou et al. [29], where polysaccharides from *Trametes robiniophila* Murr were able to negatively regulate P-gp in human hepatocellular carcinoma cells and in nude mice in vivo [29]. Another study showed that ergosterol peroxide isolated from the lipid fraction of *Ganoderma lucidum* had the ability to reverse the chemoresistance conferred by microRNA (miR-378) in tumour cells (U87 and MDA-MB-231 cells). Cancer cells that express miR-378 have more aggressive properties and become resistant to chemotherapy. The *Ganoderma lucidum* lipid fraction combined with chemotherapy (Epirubicin) in the MDA-MB-231 cell line transfected with miR-378 was subjected to treatment with Epirubicin (2.0 µg/mL), with or without the *Ganoderma lucidum* lipid fraction. The combination with the *Ganoderma lucidum* lipid fraction (0.4 µL/mL) significantly increased the sensitivity of miR-378 cells to Epirubicin [51]. These findings are also in agreement with the study by Gou et al. [29], in which *Trametes robiniophila* Murr played a modulating role, inhibiting P-gp expression through the regulation of microRNAs (miR-224-5p) [29]. In this way, strategies using isolated compounds allow for a targeted approach with greater purity and standardisation, but on the other hand strategies using extracts allow for a broad-spectrum approach, since extracts have various bioactive compounds, including polysaccharides, terpenoids and phenolic compounds, which can have synergistic effects on each other, as well as acting on different chemoresistance mechanisms [52,53].

Due to their therapeutic efficacy and safety properties, mushrooms are of great interest in nanobiotechnology as a drug delivery system [38]. One study reported the development of highly stable and biocompatible gold nanoblasts coated with β-glucan from the *Pleurotus tuber-regium* mushroom for photothermal cancer therapy. This approach demonstrated high colloidal stability in various biological media as well as in simulated gastric fluid, showed low cytotoxicity and was effective in treating MCF-7 cells (4.5 ± 0.9% viability) [54]. Polysaccharides extracted from *Agaricus bisporus* were microcapsulated to exert an immunotherapeutic effect by activating intestinal resident natural killer (NK) cells against colon cancer. *Agaricus bisporus* polysaccharides were microcapsulated in Alg/κ-carrageenan microcapsules as an oral administration system for colon cancer. The microcapsule showed superior acid stability, controlled release and thermal stability at high temperatures with higher hydrogel swelling rates in colon-mimicking pH. After NK cell activation, a significant increase in NK CD16+CD56+ cell populations was noted, which were activated and showed 74.09% cytotoxic effects against human colon cancer Caco-2 cells, stopping them in the G0/G1 phase and leading to apoptosis. Mechanistically, they acted by negatively regulating the BCL2 and TGF genes, making them a good oral delivery system for colon cancer immunotherapy [55]. These findings reinforce the therapeutic potential of mushroom bioactive compounds.

Another significant limitation of this study is that although all the included studies described the combined use of mushroom extracts with chemotherapeutic agents as synergistic or potentiating, none of them carried out a comprehensive analysis of inhibitory concentrations. For example, there was no adequate investigation into combinations of chemotherapeutic drugs and mushroom extracts using increasing concentrations of both substances in a fixed ratio, as suggested by methods such as the Chou–Talalay approach [56] and as highlighted in the studies by Duarte et al. [57] and Nunes et al. [58], which used a bioinformatics tool.

With regard to clinical trials, the limitation was the lack of studies that used mushroom extracts as adjuvants to chemotherapy as a strategy to overcome chemoresistance in cancer patients. This limitation emphasises the need for more research and highlights the gap in the scientific evidence currently available on this specific topic. But, on the other hand, a large number of clinical trials have used mushrooms to improve the side effects of chemotherapy, which reinforces the potential of mushrooms to help treat cancer and reduce side effects. For example, many groups have reported that mushrooms minimise side effects such as nausea, bone marrow suppression and anaemia [6]. Patients undergoing chemotherapy, when they additionally used mushroom derivatives, developed less severe gastrointestinal symptoms [59]. Loss of appetite, fever, cough, weakness, sweating and insomnia improved [60], as well as fatigue and quality of life in general [61]. Mushroom extracts could also improve the immune responses of patients undergoing chemotherapy [6,59]. Clinical trial data play a crucial role in translating in vitro or in vivo research into clinical application. The rigours of conducting clinical trials are widely recognised, especially involving cancer patients, and not only attest to the safety of using these mushroom extracts, but also provide evidence regarding the efficacy of these extracts in optimising cancer treatment. By analysing clinical results, such as a reduction in tumour size, improved survival rates and a reduction in side effects, it is possible to gain valuable insights into the impact of mushroom extracts on cancer treatment. This evidence-based approach contributes to a more comprehensive and scientifically supported understanding of the therapeutic potential of using extracts as an adjuvant in cancer treatment.

The studies reviewed investigated a variety of mushrooms, including *Agaricus blazei* Murill, *Reishi*, *Krestin*, *Agaricus bisporus*, *Lentinula edodes*, *Ganoderma lucidum*, *Pleurotus pulmonarius*, *Tricholoma anatolicum*, *Fomes fomentarius*, *Pleurotus tuber-regium* and *Trametes robiniophila* Murr. However, this particular collection only covers a small part of the mushroom species, which is estimated to number around 14,000 in total [6,62]. Research has established links between the cellular mechanisms through which mushroom-derived compounds act, highlighting their potential in anticancer therapy [39,40].

The diversity of this natural source is clearly yet to be investigated, particularly as a strategy for enhancing the effect of chemotherapy. In addition, mushrooms are well known for their antiviral effect and their strong immunomodulatory effect. It is well established that our immune system is a powerful tool in the fight against tumours. By studying the ways in which mushrooms can modulate the immune system, we can gain valuable information for the development of more effective anticancer therapies. This research has the potential to make significant contributions to the field of cancer treatment and improve patient outcomes [63]. In summary, the results of this meta-analysis suggest that mushroom extracts have the potential to restore sensitivity to chemotherapy in cancer cells. However, more studies are needed to investigate the effectiveness of a wider range of mushroom extracts with regard to these therapeutic properties.

## 5. Conclusions

In conclusion, the studies examined in this analysis collectively emphasise the potential of mushroom extracts as treatments for cancer in addressing resistance to chemotherapy. While there is clinical evidence on this specific subject, research conducted on various types of mushrooms and their extracts reveals encouraging outcomes in reversing chemoresistance, enhancing the sensitivity of cancer cells to chemotherapy and restraining the growth of cancer cells. These findings underline the necessity for further research and clinical trials to fully explore the capabilities of mushroom extracts in combating cancer and enhancing outcomes for cancer patients. The wide variety of mushrooms and their bioactive compounds offer possibilities for approaches to cancer treatment, with the ultimate aim of improving treatment effectiveness while minimising side effects.

## Figures and Tables

**Figure 1 cells-13-00510-f001:**
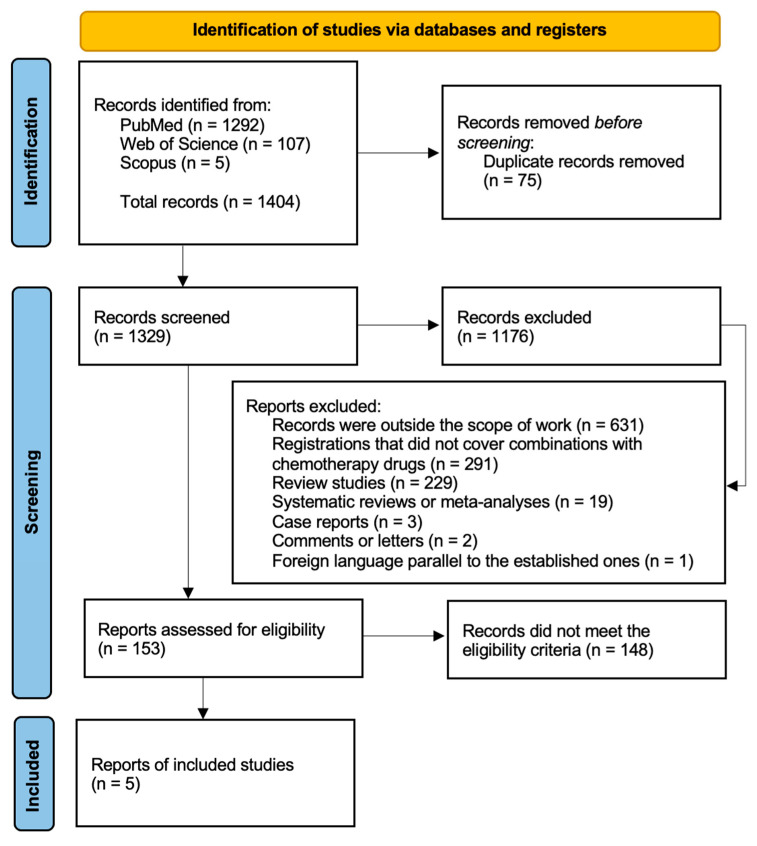
Highlighting the PRISMA flowchart for the studies included in this systematic review.

**Figure 2 cells-13-00510-f002:**
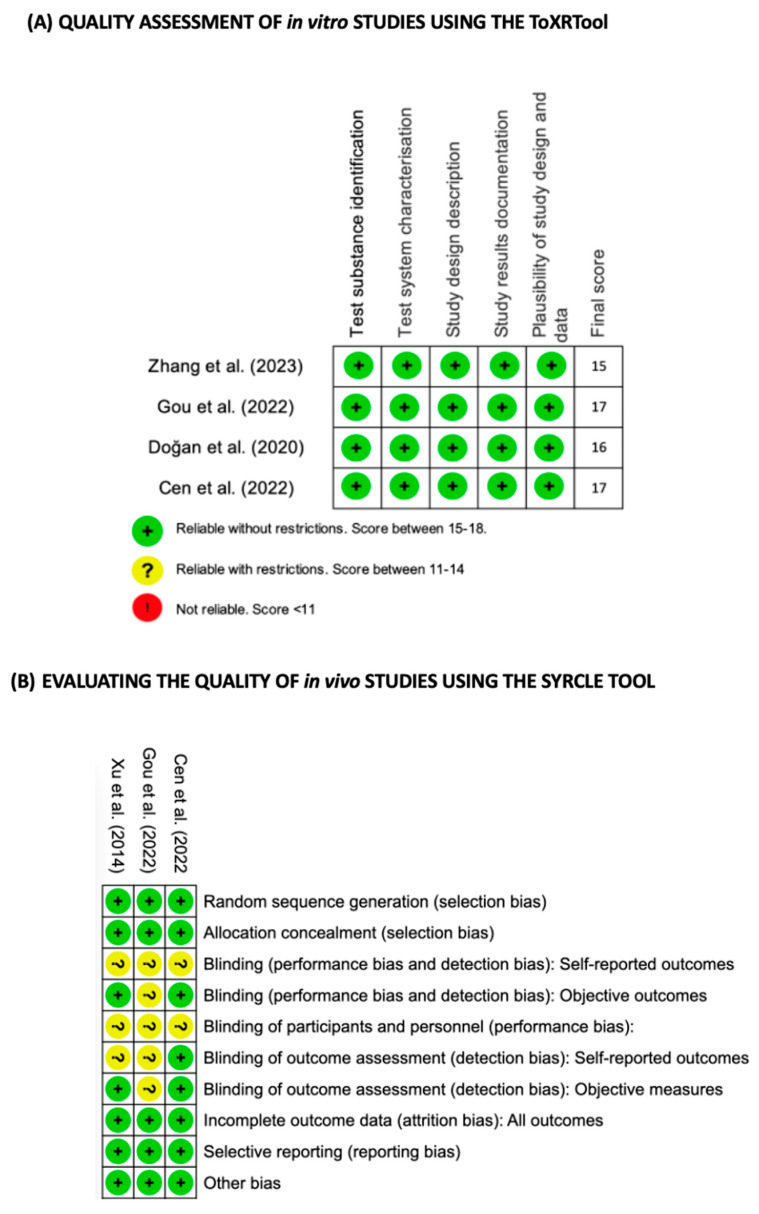
Quality assessment of in vitro and in vivo studies. (**A**) Quality assessment of in vitro studies using the toxicological data reliability assessment tool (ToxRTool). (**B**) Quality assessment of in vivo studies using the SYRCLE’s RoB tool for animal intervention studies. Red (−) = high risk of bias; Yellow (?) = unknown risk of bias; Green (+) = low risk of bias.

**Figure 3 cells-13-00510-f003:**
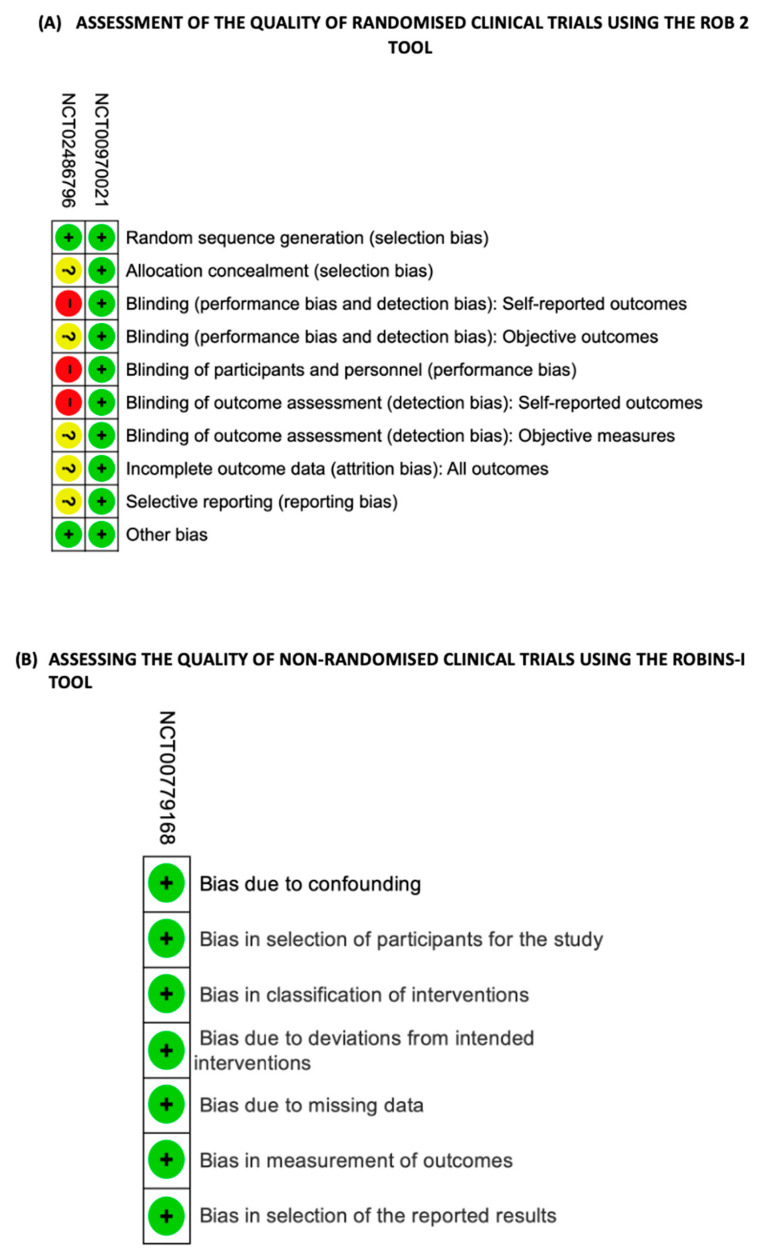
Quality assessment of randomised and non-randomised clinical trials. (**A**) Quality assessment of randomised clinical trials using the Cochrane risk-of-bias tool (RoB 2). (**B**) Quality assessment of non-randomised clinical trials using the ROBINS-I tool (Risk Of Bias In Non-randomised Studies—of Interventions). Red (−) = high risk of bias; Yellow (?) = unknown risk of bias; Green (+) = low risk of bias.

**Figure 4 cells-13-00510-f004:**
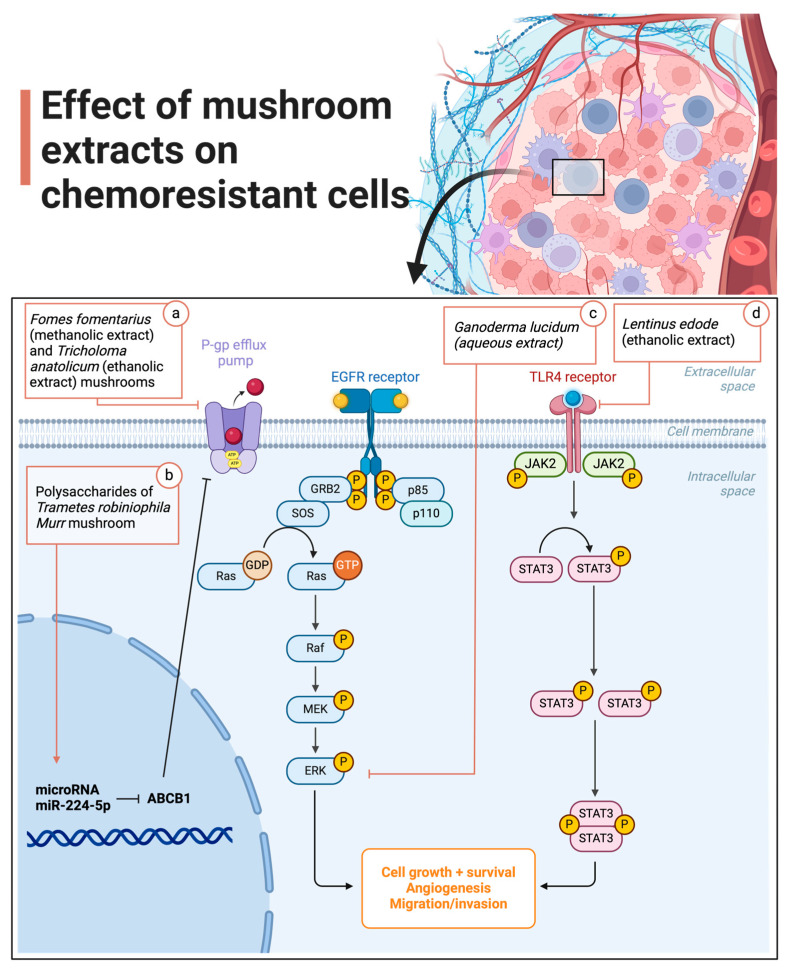
Mechanism of cellular action of mushroom extracts to improve the efficacy of chemotherapy. (**a**) Extracts of *Fomes fomentarius* and *Tricholoma anatolicum* inhibit the activity of the P-gp protein in breast cancer cells (MCF-7) resistant to vincristine and paclitaxel. (**b**) Polysaccharides extracted from *Trametes robiniophila* Murr positively regulate microRNAs (miR-224-5p), which in turn inhibit the expression of the ABCB1 gene and consequently also inhibit the expression of the P-gp protein. This increases the sensitivity of hepatoma cells to oxaliplatin treatment in human hepatocellular carcinoma cells (HepG2 and Bel-7404) and in nude mice in vivo (BALB/c). (**c**) *Ganoderma lucidum* spores, as well as Ganoderic Acid D, its main component, affect sensitivity to cisplatin by increasing intracellular reactive oxygen species (ROS) levels in tumour cells and, consequently, inhibiting signal-regulated kinase (ERK1/2) signalling in the mitogen-activated protein kinase (MAPK) pathway in SKOV3 and SKOV3/DDP (cisplatin-resistant) ovarian cancer cells. (**d**) Polysaccharides extracted from *Lentinula edodes* reverse docetaxel resistance in prostate cancer cells (PC3) induced by cancer-associated fibroblasts (CAFs) (resistant to docetaxel) by reducing the concentration of TGF-β1, which is essential for docetaxel resistance. It acts via the TLR4 receptor on CAFs, inhibiting the JAK2/STAT3 pathway, which is linked to CAF activity and TGF-β1 secretion. This deactivates CAFs and reverses docetaxel resistance in prostate cancer cells [28,29,30,31,32].

**Table 1 cells-13-00510-t001:** Description of the PICO (participants, intervention, comparison, and outcome).

Criteria	Description
Participants	Cell lines, in vitro or in vivo models of cancer with resistance to chemotherapy. Clinical studies involving humans with chemotherapy-resistant cancer.
Intervention	Mushroom extracts used in synergy or combined with anticancer therapies.
Comparison	Anticancer therapies without the use of mushroom extracts.
Outcomes	Mushroom extracts in combination with anticancer therapies to improve the efficacy of chemotherapy on chemoresistant cancer cells.
Study design	In vitro and in vivo studies and clinical trials

**Table 2 cells-13-00510-t002:** Summary of mushrooms combined with chemotherapy drugs to improve the effectiveness of chemotherapy on cancer cells.

Mushroom	Extract Type/Chemical Compound	Concentration/Dose	Type of Cancer	Model	Chemotherapy Drug	Mechanism	Ref.
*Lentinula edodes*	polysaccharide	0.25 mg/mL	prostate cancer	in vitro	docetaxel	Increased docetaxel-induced apoptosis prevents epithelial-mesenchymal transition, reduces TGF-β1 concentration and inhibits JAK2/STAT3 pathway.	[28]
*Trametes robiniophila* Murr	polysaccharides	in vitro: 0.20 to 0.27 mM in vivo: 2.6 g/kg	hepatoma	in vitro/in vivo	oxaliplatin	Inhibits P-gp expression by increasing miR-224-5p.	[29]
*Fomes fomentarius*	aqueous, methanol, and ethanol extracts	1.11–1.34 mg/mL	breast cancer	in vitro	paclitaxel and vincristine	Inhibition of P-gp activity.	[30]
*Tricholoma anatolicum*	ethanol extracts	1.26–1.81/mL	breast cancer	in vitro	paclitaxel and vincristine	Inhibition of P-gp activity.	[30]
*Pleurotus pulmonarius*	water extract	50 mg/kg	liver cancer	in vivo	cisplatin	Not described.	[31]
*Ganoderma lucidum*	water extract (Sporoderm-Broken spores)	in vitro: 200 μM in vivo: 2 g/kg	ovarian cancer	in vitro/in vivo	cisplatin	Inhibition of ERK1/2 signalling of the MAPK signalling pathway.	[32]

**Table 3 cells-13-00510-t003:** Clinical trials with mushroom extracts in combination with chemotherapeutic drugs.

Reference Number	Start of Study	Study Completion	Study Title	Conditions	Study Status	Interventions	Sample	Study Design
NCT00970021	2009-06	2014-02	*Agaricus Blazei* Murill in Patients With Multiple Myeloma	multiple myeloma	COMPLETED	Ingestion of 60 mL of placebo daily, in addition to chemotherapy, or ingestion of 60 mL of *Agaricus blazei* Murill mushroom extract daily, in addition to chemotherapy, for 7 weeks.	39	RANDOMISED
NCT00779168	2009-01	2021-06	White Button Mushroom Extract in Treating Patients With Recurrent Prostate Cancer After Local Therapy	prostate cancer	COMPLETED	*Agaricus bisporus* in staggered doses. Patients were treated with each of the following doses: 4, 6, 8, 10, 12 and 14 g of *Agaricus bisporus* powder daily.	36	NON-RANDOMISED
NCT04519879	2021-05	2024-04 (Estimated)	White Button Mushroom Sup for the Reduction of PSA in Pts With Biochemically Rec or Therapy Naive Fav Risk Prostate CA	prostate cancer	RECRUITING	Treatment with white mushroom extract orally twice a day. Treatment is repeated every 12 weeks for four cycles (48 weeks) in the absence of disease progression or unacceptable toxicity.	132	RANDOMISED
NCT02486796	2016-02	2017-03	Immediate or Delayed Naturopathic Medicine in Combination With Neo-Adjuvant Chemotherapy for Breast Cancer	breast cancer	TERMINATED	Treatment with *Reishi* mushroom extract, Coenzyme Q10 and melatonin from the first day of the first cycle of prescribed neoadjuvant chemotherapy.	5	RANDOMISED
NCT05763199	2023-12	2027-07 (Estimated)	Standardized Extract of Cultured *Lentinula Edodes* Mycelia (AHCC^®^) in Ovarian Cancer Patients on Adjuvant Chemotherapy	ovarian cancer	RECRUITING	Standardised extract of *Lentinula edodes* mycelia in culture (AHCC^®^), AHCC 3 g, orally daily or placebo daily together with chemotherapy according to the standard of treatment.	20	RANDOMISED
NCT01685489	2013-05	-	A Phase 1b Dose Escalation Trial of PSK^®^/Placebo With Docetaxel to Treat Metastatic Castration-resistant Prostate Cancer	prostate cancer	WITHDRAWN	PSK^®^ or oral placebo for 21 days, followed by treatment with standard intravenous docetaxel at 75 mg/m^2^ every 3 weeks for three cycles.	-	RANDOMISED
IRCT20190822044579N1	2021-09	-	The effect of probiotic ice cream contained ß-glucan on immune function in breast cancer patients	breast cancer	RECRUITMENT COMPLETE	Treatment with 100 g of ice cream a day for a month, containing Lactobacillus plantarum as a probiotic at a rate of at least eight logarithmic cycles per gram of ice cream, containing beta-glucan extracted from *Agaricus bisporus* (with a purity of over 70 percent) at a rate of 2 percent in the ice cream formulation, or containing Lactobacillus plantarum and beta-glucan extracted from *Agaricus bisporus*, or reception of ice cream without probiotic and beta-glucan as a placebo group.	120	RANDOMISED

## Data Availability

Not applicable.

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
