# Peer review of "The Potential of Mushroom Extracts to Improve Chemotherapy Efficacy in Cancer Cells: A Systematic Review"

_cells, 2024, doi:10.3390/cells13060510_

Round 1
Reviewer 1 Report (Previous Reviewer 1)
Comments and Suggestions for Authors
The authors have revised the manuscript based on the reviewer's previous comments, and made the necessary correction to the scientific name. Literature on the anticancer efficacy of mushrooms in animal and clinical trials has been systematically compiled. Despite that, the content of this work still does not really match the objective or main focus i.e., mushrooms and "to overcome/combat chemoresistance" (line 26, 29) - which is part of the old title, and the revised sections (highlighted) did not really address this aspect.
Comments on the Quality of English LanguageAcceptable
Author Response
Responses to the reviewers’ comments
The authors would like to thank the reviewers for their interest in this manuscript and acknowledge their valuable comments, which have increased the impact and quality of this review article. The authors have carefully reviewed the manuscript and made all the recommended changes (highlighted in yellow in the main manuscript). We are confident that the manuscript is now suitable for publication in Cells.
Comments from reviewer #1:
#1. The authors have revised the manuscript based on the reviewer's previous comments, and made the necessary correction to the scientific name. Literature on the anticancer efficacy of mushrooms in animal and clinical trials has been systematically compiled. Despite that, the content of this work still does not really match the objective or main focus i.e., mushrooms and "to overcome/combat chemoresistance" (line 26, 29) - which is part of the old title, and the revised sections (highlighted) did not really address this aspect.
Authors response:
Thank you for your feedback. We have re-evaluated the manuscript, as well as the section (highlighted), so that it is more in line with the title. Thank you for your contribution.
Comments from reviewer #2:
#1. The manuscript has been revised.
Authors response:
Thank you for the update on the manuscript revision.
Comments from reviewer #3:
#1. The authors Fonseca et al., compiled data from PubMed, Web of Science, and Scopus databases, adhering to the PRISMA 19 guidelines, and discusses the use of mushroom extracts and their metabolites in cancer therapy.
In their conclusion, a variety of mushroom extracts are utilized in medicine for treating cancer because of their low toxicity and ability to reduce adverse side effects. Clinical investigations demonstrate that they elevate p53 protein (Suppressor protein) expression, inhibit tumor development, and improve chemotherapy responsiveness, exceeding the effectiveness of individual drugs.
Further, this study explores the utilization of mushroom extracts to enhance the effectiveness of chemotherapy in overcoming chemoresistance. Despite minimal research, the variety of mushrooms and their bioactive chemicals offer potential for novel approaches to improving cancer treatment. Yet, the authors did not distinguish the extracts and varied mushrooms with unambiguous drawings. The Authors meticulously examined available literature and presented a comprehensive analysis of the current benefits and drawbacks. The text is well-written in most sections, however, there are instances of content redundancy encountered throughout. Authors should provide valuable suggestions to researchers regarding cancer therapy and drug development to enhance their work.
Authors response:
Thank you for your feedback. Thank you for your positive comments on our review. Working to minimize redundant content. And the existing design has been corrected to better distinguish the variability of mushroom extracts.
Comments from reviewer #3:
#1. The manuscript has been significantly improved and now warrants publication in this journal.
Authors response:
Thank you for your positive assessment. We're pleased to hear that the manuscript has undergone significant improvements and is now considered suitable for publication in this journal.
Reviewer 2 Report (Previous Reviewer 2)
Comments and Suggestions for Authors
The manuscript has been revised.
Comments on the Quality of English LanguageThe manuscript has been revised.
Author Response
Responses to the reviewers’ comments
The authors would like to thank the reviewers for their interest in this manuscript and acknowledge their valuable comments, which have increased the impact and quality of this review article. The authors have carefully reviewed the manuscript and made all the recommended changes (highlighted in yellow in the main manuscript). We are confident that the manuscript is now suitable for publication in Cells.
Comments from reviewer #1:
#1. The authors have revised the manuscript based on the reviewer's previous comments, and made the necessary correction to the scientific name. Literature on the anticancer efficacy of mushrooms in animal and clinical trials has been systematically compiled. Despite that, the content of this work still does not really match the objective or main focus i.e., mushrooms and "to overcome/combat chemoresistance" (line 26, 29) - which is part of the old title, and the revised sections (highlighted) did not really address this aspect.
Authors response:
Thank you for your feedback. We have re-evaluated the manuscript, as well as the section (highlighted), so that it is more in line with the title. Thank you for your contribution.
Comments from reviewer #2:
#1. The manuscript has been revised.
Authors response:
Thank you for the update on the manuscript revision.
Comments from reviewer #3:
#1. The authors Fonseca et al., compiled data from PubMed, Web of Science, and Scopus databases, adhering to the PRISMA 19 guidelines, and discusses the use of mushroom extracts and their metabolites in cancer therapy.
In their conclusion, a variety of mushroom extracts are utilized in medicine for treating cancer because of their low toxicity and ability to reduce adverse side effects. Clinical investigations demonstrate that they elevate p53 protein (Suppressor protein) expression, inhibit tumor development, and improve chemotherapy responsiveness, exceeding the effectiveness of individual drugs.
Further, this study explores the utilization of mushroom extracts to enhance the effectiveness of chemotherapy in overcoming chemoresistance. Despite minimal research, the variety of mushrooms and their bioactive chemicals offer potential for novel approaches to improving cancer treatment. Yet, the authors did not distinguish the extracts and varied mushrooms with unambiguous drawings. The Authors meticulously examined available literature and presented a comprehensive analysis of the current benefits and drawbacks. The text is well-written in most sections, however, there are instances of content redundancy encountered throughout. Authors should provide valuable suggestions to researchers regarding cancer therapy and drug development to enhance their work.
Authors response:
Thank you for your feedback. Thank you for your positive comments on our review. Working to minimize redundant content. And the existing design has been corrected to better distinguish the variability of mushroom extracts.
Comments from reviewer #3:
#1. The manuscript has been significantly improved and now warrants publication in this journal.
Authors response:
Thank you for your positive assessment. We're pleased to hear that the manuscript has undergone significant improvements and is now considered suitable for publication in this journal.
Reviewer 3 Report (Previous Reviewer 3)
Comments and Suggestions for Authors
The authors Fonseca et al., compiled data from PubMed, Web of Science, and Scopus databases, adhering to the PRISMA 19 guidelines, and discusses the use of mushroom extracts and their metabolites in cancer therapy.
In their conclusion, a variety of mushroom extracts are utilized in medicine for treating cancer because of their low toxicity and ability to reduce adverse side effects. Clinical investigations demonstrate that they elevate p53 protein (Suppressor protein) expression, inhibit tumor development, and improve chemotherapy responsiveness, exceeding the effectiveness of individual drugs.
Further, this study explores the utilization of mushroom extracts to enhance the effectiveness of chemotherapy in overcoming chemoresistance. Despite minimal research, the variety of mushrooms and their bioactive chemicals offer potential for novel approaches to improving cancer treatment. Yet, the authors did not distinguish the extracts and varied mushrooms with unambiguous drawings. The Authors meticulously examined available literature and presented a comprehensive analysis of the current benefits and drawbacks. The text is well-written in most sections, however, there are instances of content redundancy encountered throughout. Authors should provide valuable suggestions to researchers regarding cancer therapy and drug development to enhance their work.
Author Response
Responses to the reviewers’ comments
The authors would like to thank the reviewers for their interest in this manuscript and acknowledge their valuable comments, which have increased the impact and quality of this review article. The authors have carefully reviewed the manuscript and made all the recommended changes (highlighted in yellow in the main manuscript). We are confident that the manuscript is now suitable for publication in Cells.
Comments from reviewer #1:
#1. The authors have revised the manuscript based on the reviewer's previous comments, and made the necessary correction to the scientific name. Literature on the anticancer efficacy of mushrooms in animal and clinical trials has been systematically compiled. Despite that, the content of this work still does not really match the objective or main focus i.e., mushrooms and "to overcome/combat chemoresistance" (line 26, 29) - which is part of the old title, and the revised sections (highlighted) did not really address this aspect.
Authors response:
Thank you for your feedback. We have re-evaluated the manuscript, as well as the section (highlighted), so that it is more in line with the title. Thank you for your contribution.
Comments from reviewer #2:
#1. The manuscript has been revised.
Authors response:
Thank you for the update on the manuscript revision.
Comments from reviewer #3:
#1. The authors Fonseca et al., compiled data from PubMed, Web of Science, and Scopus databases, adhering to the PRISMA 19 guidelines, and discusses the use of mushroom extracts and their metabolites in cancer therapy.
In their conclusion, a variety of mushroom extracts are utilized in medicine for treating cancer because of their low toxicity and ability to reduce adverse side effects. Clinical investigations demonstrate that they elevate p53 protein (Suppressor protein) expression, inhibit tumor development, and improve chemotherapy responsiveness, exceeding the effectiveness of individual drugs.
Further, this study explores the utilization of mushroom extracts to enhance the effectiveness of chemotherapy in overcoming chemoresistance. Despite minimal research, the variety of mushrooms and their bioactive chemicals offer potential for novel approaches to improving cancer treatment. Yet, the authors did not distinguish the extracts and varied mushrooms with unambiguous drawings. The Authors meticulously examined available literature and presented a comprehensive analysis of the current benefits and drawbacks. The text is well-written in most sections, however, there are instances of content redundancy encountered throughout. Authors should provide valuable suggestions to researchers regarding cancer therapy and drug development to enhance their work.
Authors response:
Thank you for your feedback. Thank you for your positive comments on our review. Working to minimize redundant content. And the existing design has been corrected to better distinguish the variability of mushroom extracts.
Comments from reviewer #3:
#1. The manuscript has been significantly improved and now warrants publication in this journal.
Authors response:
Thank you for your positive assessment. We're pleased to hear that the manuscript has undergone significant improvements and is now considered suitable for publication in this journal.
Reviewer 4 Report (Previous Reviewer 4)
Comments and Suggestions for Authors
The manuscript has been significantly improved and now warrants publication in this journal.
Author Response
Responses to the reviewers’ comments
The authors would like to thank the reviewers for their interest in this manuscript and acknowledge their valuable comments, which have increased the impact and quality of this review article. The authors have carefully reviewed the manuscript and made all the recommended changes (highlighted in yellow in the main manuscript). We are confident that the manuscript is now suitable for publication in Cells.
Comments from reviewer #1:
#1. The authors have revised the manuscript based on the reviewer's previous comments, and made the necessary correction to the scientific name. Literature on the anticancer efficacy of mushrooms in animal and clinical trials has been systematically compiled. Despite that, the content of this work still does not really match the objective or main focus i.e., mushrooms and "to overcome/combat chemoresistance" (line 26, 29) - which is part of the old title, and the revised sections (highlighted) did not really address this aspect.
Authors response:
Thank you for your feedback. We have re-evaluated the manuscript, as well as the section (highlighted), so that it is more in line with the title. Thank you for your contribution.
Comments from reviewer #2:
#1. The manuscript has been revised.
Authors response:
Thank you for the update on the manuscript revision.
Comments from reviewer #3:
#1. The authors Fonseca et al., compiled data from PubMed, Web of Science, and Scopus databases, adhering to the PRISMA 19 guidelines, and discusses the use of mushroom extracts and their metabolites in cancer therapy.
In their conclusion, a variety of mushroom extracts are utilized in medicine for treating cancer because of their low toxicity and ability to reduce adverse side effects. Clinical investigations demonstrate that they elevate p53 protein (Suppressor protein) expression, inhibit tumor development, and improve chemotherapy responsiveness, exceeding the effectiveness of individual drugs.
Further, this study explores the utilization of mushroom extracts to enhance the effectiveness of chemotherapy in overcoming chemoresistance. Despite minimal research, the variety of mushrooms and their bioactive chemicals offer potential for novel approaches to improving cancer treatment. Yet, the authors did not distinguish the extracts and varied mushrooms with unambiguous drawings. The Authors meticulously examined available literature and presented a comprehensive analysis of the current benefits and drawbacks. The text is well-written in most sections, however, there are instances of content redundancy encountered throughout. Authors should provide valuable suggestions to researchers regarding cancer therapy and drug development to enhance their work.
Authors response:
Thank you for your feedback. Thank you for your positive comments on our review. Working to minimize redundant content. And the existing design has been corrected to better distinguish the variability of mushroom extracts.
Comments from reviewer #3:
#1. The manuscript has been significantly improved and now warrants publication in this journal.
Authors response:
Thank you for your positive assessment. We're pleased to hear that the manuscript has undergone significant improvements and is now considered suitable for publication in this journal.
This manuscript is a resubmission of an earlier submission. The following is a list of the peer review reports and author responses from that submission.
Round 1
Reviewer 1 Report
Comments and Suggestions for Authors
This manuscript deals with an interesting but extensively reviewed subject matter (=anticancer effect of mushrooms) in the field of medicinal mushrooms. While I appreciate the systematic review approach used for this work, the approach used towards data analysis is rather weak. This review appears more like a compilation of summary of selected published papers on the anticancer effect of mushrooms without a thorough and critical evaluation of the published information.
This review focused on anticancer activity/mechanism of mushroom extracts instead of chemoresistance. The title is, therefore, inaccurate. The same goes for the content of the abstract and conclusion sections which are inconsistent with the title.
It is also unclear how data from clinical trials can be used to justify the use of mushroom extracts to overcome chemoresistance.
There is a serious lack of critical discussion on the use of extracts, even though there are many published work.
Extracts are a mixture of various compounds. In the case of mushroom extracts, some bioactive components of the water and solvent extracts have been successfully isolated and characterised but this was insufficiently discussed.
The other issue is that the composition of even the same species can vary depending on factors like extraction procedure, cultivation condition, harvesting stages, etc. Has this been taken into consideration when preparing this review?
This points to the limitations when using extracts, and when making comparison on the data obtained using extracts. As such, the many work on crude extracts need to be supported with chemical profiles, or to use standardised extracts, however, none of these was mentioned and highlighted in this work.
There are numerous mistakes when it comes to the mushrooms' nomenclature. For instance, it is Lentinula edodes, NOT Lentinus edodes. The authors are suggested to seek the input of a mycologist to look through the manuscript. Furthermore, the standard way to write scientific names in manuscripts should be followed, i.e., to spell in full once and then use the abbreviated forms.
The conclusion deals with mostly the general aspects of anticancer effect of mushrooms, rather than chemoresistance, as claimed in the title.
Comments on the Quality of English LanguageAcceptable
Reviewer 2 Report
Comments and Suggestions for Authors
1. In the title and main text of the manuscript, the authors used the word "chemoresistance". Please explain in detail the meaning of this word.
2. Mushroom extracts typically contain polysaccharides, which frequently exhibit immunomodulatory activity. What is the function of polysaccharides in addressing chemical resistance?
3. The author mentioned several mushroom extracts in clinical research. What other mushroom extracts with chemical resistance activity have not yet been utilized in clinical practice and are being researched? Please consider listing in a table.
Comments on the Quality of English LanguagePlease further improve the fluency and readability of the manuscript.
Reviewer 3 Report
Comments and Suggestions for Authors
To the authors
The authors comprehensively reviewed and evaluated the cellular mechanism by which mushroom extracts can combat chemo-resistance. In addition to having relevant material and a suitable PICO description, the systematic review is well-structured and adequately expressed. In addition, the Rayyan AI tool was designed for conducting systematic literature reviews. It helps researchers sift through many academic papers to identify relevant studies for a particular research question or topic.
On the other hand, the authors of the review article highlighted the Agaricus species sixteen times, but Ganoderma was only highlighted five times. Compared to Agaricus species, which are mainly used for their medicinal and pharmacological properties, Ganoderma species have superior therapeutic and pharmacological properties.
Reviewer 4 Report
Comments and Suggestions for Authors
The manuscript: Cellular mechanism of action of mushroom extracts to tackle chemoresistance: A Systematic Review, describes an interesting paper to look for an alternative approach in the treatment of cancer. The ms is well described and understandable, but the authors only describe a few species of mushrooms, and suggest the applications indicated there. They should consider this detail in the tittle of the paper, because they do not know how the great diversity of these organisms present the activity in the treatment of cancer.